# Role of *ABCA1* in Cardiovascular Disease

**DOI:** 10.3390/jpm12061010

**Published:** 2022-06-20

**Authors:** Jing Wang, Qianqian Xiao, Luyun Wang, Yan Wang, Daowen Wang, Hu Ding

**Affiliations:** 1Division of Cardiology, Department of Internal Medicine, Tongji Hospital, Tongji Medical College, Huazhong University of Science and Technology, Wuhan 430030, China; wangjing81531@163.com (J.W.); xiaoqq9711@163.com (Q.X.); wangluyun2004@126.com (L.W.); newswangyan@tjh.tjmu.edu.cn (Y.W.); dwwang@tjh.tjmu.edu.cn (D.W.); 2Hubei Key Laboratory of Genetics and Molecular Mechanisms of Cardiological Disorders, Wuhan 430030, China

**Keywords:** cardiovascular disease, inflammation, polymorphism, post-translational modification (PTM), transcription regulation, ATP binding cassette transporter 1 (*ABCA1*), cholesterol, high density lipoprotein cholesterol (HDL-C)

## Abstract

Cholesterol homeostasis plays a significant role in cardiovascular disease. Previous studies have indicated that ATP-binding cassette transporter A1 (*ABCA1*) is one of the most important proteins that maintains cholesterol homeostasis. *ABCA1* mediates nascent high-density lipoprotein biogenesis. Upon binding with apolipoprotein A-I, *ABCA1* facilitates the efflux of excess intracellular cholesterol and phospholipids and controls the rate-limiting step of reverse cholesterol transport. In addition, ABCA1 interacts with the apolipoprotein receptor and suppresses inflammation through a series of signaling pathways. Thus, *ABCA1* may prevent cardiovascular disease by inhibiting inflammation and maintaining lipid homeostasis. Several studies have indicated that post-transcriptional modifications play a critical role in the regulation of ABCA1 transportation and plasma membrane localization, which affects its biological function. Meanwhile, carriers of the loss-of-function *ABCA1* gene are often accompanied by decreased expression of *ABCA1* and an increased risk of cardiovascular diseases. We summarized the *ABCA1* transcription regulation mechanism, mutations, post-translational modifications, and their roles in the development of dyslipidemia, atherosclerosis, ischemia/reperfusion, myocardial infarction, and coronary heart disease.

## 1. Introduction

According to the World Health Organization, cardiovascular diseases are the leading cause of mortality [1], causing a huge financial burden on the society [2]. Approximately 17.9 million people died from cardiovascular diseases in 2019, accounting for 32% of global deaths. Of these, 85% died from heart attack and stroke. Additionally, with the aging of the global population [3], the United Nations predicted that nearly one in six individuals would be over the age of 65 by 2050 [4]. As previously reported, cardiovascular diseases cause an enormous burden on elderly patients [5] and significantly affect their quality of life [6]. Cardiovascular diseases involve the heart and blood vessels. Many factors contribute to the development and progression of cardiovascular diseases, including sex, age, poor diet, exercise, obesity, smoking, alcohol consumption, high cholesterol, hypertension, diabetes, and other psychosocial factors. Moreover, clinical trials and genetic epidemiological studies have shown that high-density lipoprotein cholesterol (HDL-C) is a clinically valuable predictor of cardiovascular disease risk instead of an independent risk factor [7].

The *ABC* transporter super-family is a large class of transmembrane proteins that bind ATP and use its energy to drive the transport of various substrates across cell membranes, including metabolites, lipids, cytotoxins, and drugs [8,9]. The current human genome annotation presents 49 *ABC* genes, which are arranged in seven subfamilies designated ‘*A*’ to ‘*G*’ [10]. As reported in the literature, there are many commonalities among *ABC* transporter super-family members, such as their material transport function and structure. The highest expression of these genes is found in critical barriers such as the placental barrier [11], blood-brain barrier [12,13], and the venous endothelium [14,15]. Among them, ATP-binding cassette transporter A1 (*ABCA1*) is the most widely studied gene and is also most closely associated with plasma high-density lipoprotein (HDL) levels.

In this review, we used cardiovascular disease; inflammation; polymorphism; post-translational modification (PTM); transcription regulation; ATP binding cassette transporter 1 (*ABCA1*); cholesterol; high density lipoprotein cholesterol (HDL-C) as keywords to search interested literatures. In this paper, the literatures on *ABCA1* from 1975 to 2022 were reviewed, most of which were in the last 10 years. *ABCA1* is located on chromosome 9q31.1. The length of the *ABCA1* gene sequence is 149 kb and it contains 50 exons and 49 introns. ABCA1 is a 254 kD integral membrane protein composed of 2261 amino acids [9]. *ABCA1* is expressed in various tissues, including the liver [16], intestine [17], placenta [18], pancreas [19], lung [20], and heart [21]. It is also expressed in macrophages [22] and endothelial cells [23]. It participates in numerous physiological and pathological processes [24], including inflammation [25], cancer development [26], dysregulation of lipid metabolism [27], type 2 diabetes mellitus [28], and cardiovascular diseases [29]. Although the human *ABCA1* gene was cloned in 1994, its biological function was not determined until 1999. Therefore, better understanding *ABCA1* is particularly important to the development of drugs based on this gene, especially those aimed at targeting cholesterol deposits in artery vessels.

Moreover, cholesterol is the major risk factor for cardiovascular disease developing processes [30]. Several studies have indicated that cholesterol is an essential biomolecule involved in multiple cellular and systemic functions [31]. Cholesterol dysregulation is a pivotal risk factor and a likely causal agent of cardiovascular diseases [32]. Previous studies of ABCA1 have revealed that it mainly participates in cholesterol efflux and binds to apolipoprotein A-I (ApoA-I) in nascent HDL formation. Reverse cholesterol transport (RCT) is defined as the movement of excess cholesterol from the peripheral tissues to the liver for biliary excretion [33]. Moreover, it is widely acknowledged that HDLs work as “good cholesterol” with atheroprotective function [34]. Thus, impaired *ABCA1* function may critically influence cholesterol homeostasis, nascent HDL biogenesis, and RCT. *ABCA1* plays a pivotal role in maintaining cholesterol homeostasis and has biomedical significance in protecting against cardiovascular disease. This review summarizes the current knowledge on *ABCA1* transcription regulation mechanism, gene polymorphism, post-translational modification, and its role in the development of diverse cardiovascular diseases, highlighting *ABCA1* as a potential therapeutic target for cardiovascular diseases.

## 2. Transcription Regulation of the *ABCA1* Gene

*ABCA1* is a key transporter that mediates cholesterol efflux from cells and is the most studied member of the *ABC* superfamily. According to the literature, the *ABCA1* gene can be regulated in multiple ways. The most common regulatory mechanism involves the transcription factors interacting with the upstream transcription initiation site to activate or inhibit *ABCA1* expression (Figure 1). In addition, many signaling molecules are involved in *ABCA1* regulation. Previous studies have reported that *ABCA1* is a target gene of the nuclear receptor superfamily, including—but not limited to—the liver X receptor (LXR) [35], retinoid X receptor (RXR) [36], retinoic acid receptor [37,38], and peroxisome proliferator-activated receptor gamma (PPAR-γ) [39]. These nuclear receptors mainly upregulate *ABCA1* expression by binding to the four-nucleotide (DR-4) element of the *ABCA1* promoter [40]. It should be noted that although the elevation of cyclic adenosine monophosphate [41] (*cAMP*) levels increases *ABCA1* expression, the response element for cAMP in the existing promoter sequences or the precise regulatory mechanism of *ABCA1* are still unclear. Some negative transcription factors downregulate *ABCA1* expression. For example, activator protein 2 (AP2) [42] interacts with the AP2-binding site in the *ABCA1* promoter region and sterol regulatory element-binding protein 2 (SREBP2) [43] and upstream stimulation factor interact with the E-box binding element [27,44]. Moreover, C-X-C motif chemokine ligand 12 (CXCL12) downregulates *ABCA1* expression by inhibiting the binding of transcription factor 21 (TCF21) to the *ABCA1* promoter [41] (Figure 1).

## 3. *ABCA1* Gene Mutation and Single Nucleotide Polymorphism

The *ABCA1* gene is a 147.2 kb DNA segment located on 9q31.1. The full-length *ABCA1* mRNA is 10,412 nt in length and has 50 exons. At least 50 types of *ABCA1* mutations have been identified, including 23 missense mutations, 6 nonsense mutations, and 21 insertion or deletion mutations [45]. Most mutations resulted in a reduction in lipid efflux. For example, a homozygous defect of the *ABCA1* gene is the molecular basis of Tangier disease (TD) [46,47]. Familial hypo-alpha-lipoproteinemia is characterized by severe HDL deficiency and premature atherosclerosis. *ABCA1* exerts a rate-controlling step in HDL biogenesis [48]. The early onset of atherosclerotic cardiovascular disease (ASCVD) is often associated with reduced HDL cholesterol levels [49,50].

Genome-wide association studies (GWAS) have identified many functional SNPs located in *ABCA1* that are associated with cardiovascular diseases. The-565C > T polymorphism in the *ABCA1* gene promoter region was associated with not only changes in *ABCA1* expression but also atherosclerosis severity [51]. Moreover, four *ABCA1* promoter SNPs have been reported to significantly influence HDL concentration. Previous studies have demonstrated that the G-395C, C-290T, C-7T [52], and -14 > T [53] polymorphisms have a significant impact on HDL. Among them, G-395C, C-290T, and C-7T were reported to be negatively related to serum HDL levels, -14 > T positively correlated with HDL levels, and variations in the *ABCA1* non-coding regions G-191C, C69T, C-17G, and InsG319 closely related to clinical outcomes but did not alter serum lipid levels in coronary artery disease (CAD) patients [54]. In the 5′ fragment of the *ABCA1*, -477C/T polymorphism showed a strong association with the severity of coronary atherosclerosis and a moderate association with serum HDL-C and ApoA-I levels [55].

Studies on SNPs in the *ABCA1* coding region have shown different associations between plasma lipid levels and coronary heart disease (CHD) susceptibility. The rs2230806 (R219K), rs2066718 (V771M), and rs4149313 (M8831I) polymorphisms (patients with GG, AA, and GG genotypes, respectively) were associated with a protective role for CHD. However, the rs9282541 (R230C) T allele increases the risk for the advancement of CHD [56,57,58,59]. Moreover, R219K and M8831I variants are associated with HDL-C elevation and triglyceride reduction [60]. Variants of V771M increased both HDL-C and ApoA-I levels. These results indicate that *ABCA1* gene polymorphism may serve as a risk or protective indicator of cardiovascular diseases. Further studies are needed to explore the impact of *ABCA1* polymorphisms on plasma lipid profiles and cardiovascular diseases.

## 4. Association of Gene Polymorphism with Cardiovascular Risk in Different Ethnicities and Sexes

It is worth noting that *ABCA1* gene polymorphism differs among different ethnicities. For example, the results of a stratified analysis by ethnicity showed that R219K polymorphism is significant associated with East Asians and other populations, but not with Caucasians [61]. A number of studies indicate that the R219K polymorphism of *ABCA1* is a protective factor for developing CHD [62,63,64]. Doosti et al. [65] reported that the presence of the GG genotype of R219K in Iranians increases their susceptibility to CAD development [65]. Similarly, *ABCA1* (R219K) gene polymorphism is closely associated with the risk of premature CAD in Egyptians [66]. Additionally, it is well established that major differences exist in the development of cardiovascular diseases between men and women, such as symptoms, epidemiology, pathophysiology, treatment, and clinic outcome [67,68,69]. Several longitudinal epidemiological studies indicate that the risk of cardiovascular disease is significantly greater in women with low estrogen levels [67,70,71,72]. Until now, studies on sex differences in the risk of cardiovascular diseases have mostly focused on the effects of sex hormones [73]. Kolovou et al. [74] report that the KK genotype of the R1587K *ABCA1* gene presented lower lipoprotein cholesterol (LDL-C) levels in a Greek female population [74]. There are many potential mechanisms for this sex difference. These include genetic mechanisms, epigenetic mechanisms, sex hormones and sex hormone receptors, and sex differences in biological processes in cardiovascular cells. Based on this, further studies are needed to explore the possible mechanisms underlying ethnicities/sex differences in cardiovascular diseases and more precise treatment in personalized medicine.

## 5. Protective Polymorphism Related to the *APOA-I* Pathway

Several studies have identified some protective polymorphisms in the *ABCA1* gene related to the *APOA-I* pathway. Delgado-Lista et al. [75] report that APOA-I levels of the major allele homozygotes of *ABCA1* single nucleotide polymorphism i48168 and i27943 are high [75]. Similarly, homozygotes of the K219 allele also have higher serum HDL-C and APOA-I levels than carriers of the R219 allele [76]. However, Zhao L. et al. [77] show that both RR and RK genotypes of the R219K *ABCA1* gene have high APOA-I levels in abdominal aortic aneurysm patients [77]. Two polymorphisms of the *ABCA1* gene, C-564T and R1587K, are related to the serum levels of APOA-I [78]. Intriguingly, there were two damaging mutations in the APOA-I gene that decrease with APOA-I production [79]. Because APOA-I is central to HDL production and RCT, it is important to further explore protective polymorphism.

## 6. ABCA1 Protein

### 6.1. *ABCA1* Structure and Distribution

ABCA1 is a 254 kD membrane transporter protein composed of 2261 amino acids. The ABCA1 molecule contains two symmetrical transmembrane domains, each of which consists of six transmembrane segments and one nucleotide binding domain (NBD) repetitive sequence [46,80]. Additionally, ABCA1 has two large extracellular domains (ECDs) [81,82] and a highly conserved N-terminal 40 amino acids sequence. N-linked glycosylation sites are common in the ABCA1 protein and seven glycosylation sites on the ECDs were successfully resolved [46]. In addition, many other modification sites exist in ABCA1, such as ubiquitylation, phosphorylation, lipidation, and palmitoylation sites. In human organs, *ABCA1* has low tissue specificity. It is highly expressed in the liver, placenta, small intestine, and lungs. At the cellular level, ABCA1 is the most abundant protein in inflammatory cells, especially macrophages.

Qian et al. [46] report on the discovery of the cryo-EM structure of human ABCA1. The researchers first analyze the single particle cryo-EM structure of the full-length human ABCA1 protein, which has an overall structure of 4.1 Å nominal resolutions and 3.9 Å for the ECD. Contrary to previous reports, this study reveals, for the first time, that the nucleotide-binding domain of ABCA1 exhibits an “outward-facing” conformation rather than an “inward-facing” conformation. Additionally, the extracellular region of ABCA1 forms a specific unique structure containing an elongated hydrophobic tunnel, which provides a key clue for further functional studies. In summary, analysis of the ABCA1 EM structure not only establishes an important foundation for understanding its functional role and the pathogenesis of related diseases, but it also expands our understanding of the plausible mechanism of transmembrane transporters.

### 6.2. *ABCA1* Post-Translational Modifications

The term ‘post-translational modifications’ (PTMs) refers to the chemical modification of proteins. These include changes in protein structure, spatial orientation, activity, stability, localization, and interactions. Thus, PTMs are at the core of many cellular signal processing events [83]. Protein PTMs have been reported to be involved in the functional expression of ABC transporters through a wide range of molecular mechanisms. ABC superfamily protein PTMs are crucial for their biological functions, such as the distribution, excretion, and up-take of endogenous compounds and xenobiotics [84]. To date, there are 461 different types of PTMs in the UniProt database for eukaryotic proteins [85]. There are various chemical modification sites in ABCA1. The most common are glycosylation, ubiquitination, phosphorylation, and palmitoylation (Figure 2).

#### 6.2.1. ABCA1 Glycosylation

Protein glycosylation refers to the election of target protein amino acid residues by covalent attachment mono-sugars or glycans, i.e., multi-sugar polysaccharides or complex oligosaccharides [87]. It is one of the most common types of PTM. To date, several different types of protein glycosylation have been reported, including N-glycosylation [88,89], O-glycosylation [90,91], C-glycosylation [92,93], S-glycosylation [94,95], and P-glycosylation [87].

Glycosylation mainly occurs in the endoplasmic reticulum (ER) and the Golgi apparatus. N-Glycosylation is the most common type of glycosylation in eukaryotes. Moreover, the ABCA1 glycosylation sites were mostly located in the ECDs. Previously, it was reported that the N-linked glycosylation sites of ABCA1 were located in the asparagine residue (N) [80]. Based on data analysis from the Uniprot/SwissProt protein database, 21 putative N-glycosylation sites were predicted in the ABCA1 amino acid sequence. To date, 7 of the 21 sites, N98, N400, N489, N521, N1453, N1504, and N1637 have been located [46,96,97,98]. However, the biological role of glycosylation of ABCA1 has not been fully elucidated yet.

As mentioned above, the R587W and Q597R mutations in *ABCA1* protect against digestion by the PNG enzyme (PNGase), which makes it less susceptible to glycosylation. These two mutations appear to be associated with TD [97]. Appropriate glycosylation of ABCA1 in ECD1 is critical for maintaining the balance of serum HDL-C levels. Previous studies have identified that Nef-mediated inactivation of ABCA1 leads to cholesterol accumulation and augmentation of lipid raft abundance, thereby increasing the risk of atherosclerosis. It is worth noting that Nef interacts with the ER chaperone calnexin to regulate glycosylation, protein folding and maturation [99]. As previously highlighted, N-acetylglucosaminyltransferase V (GnT-V) is an important glycosyltransferase. Interestingly, GnT-V can significantly increase ABCA1 expression and cause aberrant glycosylation of HDL-C assembly [100], which suggests that glycosylated modification of ABCA1 is essential for its biological functions in HDL production and lipid homeostasis. The mechanism by which ABCA1 glycosylates remains unclear. Here, we outline several investigations that provide novel insights into ABCA1 glycosylation modification and the risk of cardiovascular diseases. Further studies are needed to explore the function of different glycosylated residues in ABCA1 and their precise mechanisms.

#### 6.2.2. ABCA1 Ubiquitination

The ubiquitin system was first discovered in 1975 [101]. Subsequently, numerous studies have confirmed that the ubiquitin-proteasome system (*UPS*) controls a wide range of cellular functions and plays a critical role in maintaining homeostasis of the body [102,103,104,105]. For the most part, the *UPS* degrades intracellular proteins through the non-lysosomal pathway [106,107,108].

Protein ubiquitination is critical for several pathophysiological processes. The role of ubiquitination in the development of cardiovascular diseases has been reviewed in previous studies [109,110,111]. A previous study discovered that ubiquitin involves ABCA1 protein proteolysis through the lysosomal and non-lysosomal degradation pathways [112]. To date, many studies have indicated that cell surface-resident ABCA1 (csABCA1) is ubiquitinated and subsequently lysosomally degraded [113]. E3 ubiquitin ligase is also involved in ABCA1 degradation [114]. So far, we have summarized diverse ways to regulate ABCA1 protein levels through the ubiquitin–proteasome pathway.

Interestingly, under conditions of cellular cholesterol accumulation, the isolation of LXRβ from csABCA1 promotes its ubiquitination [113]. However, in CHO cell lines, ubiquitination of ABCA1 was decreased by cell cholesterol loading [115]. Meanwhile, activation of the endosomal sorting complex required for transport (*ESCRT*) pathway increased the degradation of csABCA1 [116]. The long form of serine/threonine kinase, a proto-oncogene, promotes the interaction between csABCA1 and LXRβ, thereby protecting against ubiquitination and degradation through the *ESCRT* system [117]. Subunit CSN8 of the COP9 signalosome controls the ubiquitinylation and deubiquitinylation of ABCA1 [118]. It has been reported that overexpression of *COP9* signalosome subunit 3 and *COP9* signalosome subunit 2 (*CSN2*) promotes ABCA1 deubiquitinylation and degradation [119]. Notably, the ApoA-I binding protein binds ApoA-I to prevent ABCA1 degradation by *CSN2* [120]. It has been reported that the ubiquitin-proteasome system mediates ABCA1 polyubiquitination and degradation [121]. AGE-albumin enhances ABCA1 degradation via ubiquitin–proteasome pathway [122]. The HIV-1 Nef protein interacts with the ABCA1 C-terminal amino acids and facilitates ABCA1 ubiquitinylation degradation via the proteasomal degradation pathway [123]. In a mouse model of ischemia-reperfusion, TANK-binding kinase 1 activation decreased ABCA1 protein levels through ubiquitinylation [124]. α-Taxilin protein deficiency aggravates ABCA1 polyubiquitination and ultimately leads to dyszoospermia [125]. Moreover, the HECT domain E3 ubiquitin protein ligase 1, an E3 ubiquitin ligase, is involved in ABCA1-mediated cholesterol export from macrophages [114]. Pulmonary adenoma resistance 1 is mediated by Cullin3-based ubiquitin E3 ligase-dependent ABCA1 ubiquitination degradation [126]. It has been reported that the ubiquitin–proteasome pathway is triggered by ubiquitin interacting with the lysine residue of the substrate protein. As stated above, ABCA1 ubiquitination and degradation are induced in many context-specific ways. However, it is still difficult to clarify the precise ubiquitin-modified lysine residue in ABCA1, which requires further study. Collectively, these results indicate that ABCA1 activation is negatively regulated by the ubiquitin-dependent proteasomal degradation pathway. Thus, modulation of ABCA1 ubiquitination provides a novel therapeutic target for atherosclerosis treatment [117].

#### 6.2.3. ABCA1 Phosphorylation

Protein phosphorylation refers to the introduction of negatively-charged phosphate groups via the chemical modification of specific protein residues (e.g., Ser, Thr, Tyr, Asp, Glu, Cys, His, Lys, and Arg) [127]. This leads to changes in protein conformation and functional activities. Protein phosphorylation is an essential and reversible modulatory mechanism that participates in nearly every basic eukaryotic cellular biological process. Moreover, phosphorylation and de-phosphorylation of kinases and phosphatases can activate or inactivate many enzymes and receptors [128,129].

It is well known that phosphorylation of serine (Ser) and/or threonine (Thr) residues in amino acid residues are catalyzed by protein kinase C (PKC) and/or protein kinase A (PKA) [130]. ABCA1 phosphorylation status is closely related to its stabilization [131]. It has been reported that there are two phosphorylation sites, Ser-1042 and Ser-2054, located in the NBDs of ABCA1, both of which can be phosphorylated by PKA [132]. Furthermore, the ABCA1–PEST sequence contains two constitutively phosphorylated sites, Thr-1286 and Thr-1305 [133]. In addition, Stein et al. [134] demonstrated that ABCA1 NBD1 + R1 is phosphorylated by protein kinase 2 (CK2) and reported on its potential phosphorylation sites (Thr-1242, Thr-1243, and Ser-1255). According to a previous study, CK2 may act as an inhibitor of ABCA1 activation [134].

Interestingly, previous studies report that ApoA-I induces the phosphorylation of ABCA1 via the PKC pathway, which protects ABCA1 against calpain-mediated proteolytic degradation to stabilize ABCA1 [135]. However, it is unclear which phosphorylation site in ABCA1 is modified by PKC. 8-Br-cAMP facilitates ABCA1 phosphorylation in a time-dependent manner. H-89 PKA significantly inhibited ABCA1 through the *cAMP*/*PKA*-dependent pathway [41]. Unsaturated fatty acids phosphorylate and destabilize ABCA1 through the phospholipase D2 and *PKC**δ* signaling pathway [136,137]. However, the polyunsaturated fatty acids eicosapentaenoic acid and ApoA-I mimetic peptide mediate ABCA1 serine dephosphorylation through the *cAMP/PKA* pathway [138,139]. Berberine attenuates the ABCA1 serine residues in a time- and dose-dependent manner [140]. These findings indicate that ABCA1 phosphorylation plays a critical role in apoA-I-mediated cholesterol efflux and atherosclerosis. Therefore, the mechanism of ABCA1 phosphorylation requires further investigation.

#### 6.2.4. ABCA1 Palmitoylation

Palmitoylation of proteins, one of the most common recognized forms of fatty acylation, plays a role in regulating protein activity, stability, and localization; membrane topology; and interactions between proteins and cofactors by imparting the spatiotemporal regulation of protein hydrophobicity [141,142]. Reversible chemical ligation of cysteine residues by palmitic acid molecules in the presence of palmitoyl acyltransferase (PAT) has been identified as either protein S-palmitoylation or S-acylation [143]. Most protein palmitoylation is catalyzed by proteins of the Asp-His-His-Cys (DHHC) family, which possess PAT functions [144].

Numerous studies have shown that palmitoylation of ABCA1 is crucial for its transportation and localization to the plasma membrane [145]. Activation of ABCA1 by SPTLC1 can be removed from the ER for transport to the Golgi apparatus [146]. A major finding suggested that palmitoylation of ABCA1 occurred in four different cysteine residues of amino acid residues: C3S, C23S, C1110S, and C1111S. A variety of enzymes, including DHHC8, are involved in palmitoylation of ABCA1. Activation of DHHC8 results in palmitoylation of ABCA1, which increases its hydrophobic property and ABCA1-mediated cholesterol efflux [145]. Taken together, these studies indicate that palmitoylation of ABCA1 plays an important role in its subcellular distribution and biological functions. Therefore, further studies are needed to decipher the regulatory mechanism underlying palmitoylation of ABCA1. This may provide a novel potential therapeutic target for increasing *ABCA1* activity to reduce foam cell formation and prevent atherosclerosis development.

### 6.3. Mechanisms of ABCA1 That Regulate Cholesterol Homeostasis

RCT is the only way to eliminate excessive cholesterol, which is of great significance to maintain the homeostasis of cholesterol metabolism. The key step for RCT is ABCA1, which binds to apolipoprotein to participate in the formation of HDL [147,148]. Recently, Yu et al. reported that there are five potential mechanisms underlying the regulation of cholesterol homeostasis by ABCA1 [149], including channel trafficking [46], ABCA1 dimerization [150,151] the promotion of the efflux of intracellular cholesterol to apoA-I by ABCA1 through a two-step process [152,153], apoA-I-free vesicle [154], and retroendocytosis [155,156]. In the future, additional work is needed to precisely elucidate the underlying mechanisms by which *ABCA1* regulates cholesterol homeostasis.

## 7. *ABCA1* and Cardiovascular Diseases

### 7.1. Dyslipidemia

Dyslipidemia is defined as a variety of lipid abnormalities and is probably related to a combination of increased total triglyceride, cholesterol, and low-density LDL-C levels, or decreased HDL-C levels. Substantial epidemiological evidence suggests that dyslipidemia is a critical risk factor for the development of ASCVD [157]. Accumulating evidence suggests that dysregulation of *ABCA1* may mediate dyslipidemia. Therefore, it is important to address how *ABCA1* modulates lipid homeostasis.

*ABCA1* is a critical regulator of HDL-C biogenesis and RCT. To date, the effect of *ABCA1* on plasma HDL-C modulation is clear. As mentioned above, TD is caused by an *ABCA1* gene mutation and is characterized by a complete deficiency or extremely low levels of HDL-C [158]. Abundant evidence suggests that *ABCA1* is involved in other types of lipid regulation. A study involving 363 patients indicated that a common variant, rs2230806, of the *ABCA1* gene led to TD and affected plasma triglyceride (TG) levels compared with that in control patients [159]. Several studies have shown higher plasma TG and lower LDL-C levels in homozygous TD patients compared with normal subjects [160,161,162]. In GWAS studies, more and more SNPs in *ABCA1* loci are reported with the effect on LDL-C (rs11789603, rs2066714, rs2740488, rs7873387, and rs2575876) and TG (rs1800978, rs1799777, rs2575876, and rs1883025) levels. Moreover, several lines of evidence suggest that many pharmacological and molecular regulators participate in *ABCA1* regulation and affect lipid levels. For instance, liraglutide upregulates *ABCA1* by phosphorylating ERK1/2 to decrease TC, TG, and LDL-C [163]. Mangiferin significantly reduces serum TG, TC, and LDL-C levels by modulating *PPAR**γ–LXRα-ABCA1/G1* pathway [164]. BBR increases *ABCA1* expression by activating the *PCK**δ* pathway to reduce hepatic TC and TG levels [140]. The loss of function of ferredoxin reductase and/or *p53* represses *ABCA1* expression, leading to an accumulation of TG, TC, and lipid droplets [165]. E1231, an agonist of *sirtuin-1*, elevates LXRα-targeted *ABCA1* expression to lower plasma TG and TC levels [166]. Methyl protodioscin promotes *ABCA1* expression by inhibiting *microRNA 33a/b* and sterol regulatory element binding protein (*SREBP*) transcription to decrease TG and TC levels [167].

Several epidemiological studies have demonstrated that proper management and prevention of dyslipidemia can significantly decrease cardiovascular morbidity and mortality [157,168,169]. In recent years, increasing attention has been paid to the use of lipid-lowering drugs. Previous studies identified many novel lipid biomarkers applied to clinical treatment, such as *PCSK9* inhibitors [170], antisense oligonucleotides of apolipoprotein C3 or angiopoietin-like 3, which significantly decrease plasma TG [171,172] and lipoprotein(a) (Lp(a)) antisense oligonucleotide levels; the latter exhibits great potential in reducing Lp(a) [173]. Collectively, the loss of function of *ABCA1* leads to dyslipidemia, and currently, there are no efficient drugs targeted to lower TC and HDL-C levels. This suggests that *ABCA1* may be a potential therapeutic target for cholesterol regulation.

### 7.2. Atherosclerosis

Atherosclerosis resulting in ischemic heart disease (IHD) is a major cause of all-cause mortality [174]. Emerging evidence indicates that the rupture of atherosclerotic lesions is closely related to cardiovascular events [175,176]. It is now generally accepted that atherosclerosis is caused by the accumulation of cholesterol and triglycerides in the arterial wall [177] and is a chronic inflammatory disease [178]. Recent studies have challenged the protective effects of HDL against atherosclerosis [179,180]. *ABCA1* expression is high in atherosclerotic tissues, especially in atherosclerotic lesions containing inflammatory cells and lymphocytes [181]. Moreover, atherosclerosis in *ABCA1* transgenic and knockout mouse models was reported to increase significantly [182]. It is well known that the development of atherosclerosis lesion requires TC, TG, and LDL-C accumulations and the existence of other risk factors, including cigarette smoking, hypertension, and diabetes mellitus. Recently, dysregulation of the immune system was identified as a novel risk factor for atherosclerosis [179].

Many studies have shown that *ABCA1* may play a dual role in the development of atherosclerosis. *ABCA1* plays a crucial role in HDL-C production and cholesterol efflux thereby protecting against atherosclerosis [183]. In contrast, it can also decrease macrophage inflammation [184]. In this context, overexpressed *ABCA1* in endothelial cells (ECs) has anti-inflammatory effects and increases cholesterol efflux [23]. Moreover, the different underlying mechanisms of *ABCA1* and atherosclerosis have been illustrated in many studies [185,186]. Annexin A1 (*ANXA1*) interacts with *ABCA1* to exert its anti-atherogenic function [187]. *CXCL12* also plays a pro-atherogenic role through the *CXCR4/GSK3β/β–catenin T120/TCF21* pathway to repress *ABCA1* expression [188]. In vascular smooth muscle cells, the inhibition of myocardin regulates *ABCA1* to prevent atherosclerosis [189]. Recently, E17241 (4-(1,3-dithiolan-2-yl)-N-(3-hydroxypyridin-2-yl) benzamide) was identified as a novel *ABCA1* upregulator and reduced atherosclerotic lesion areas in vivo in animal models [190]. An in vitro study showed that mangiferin, an agonist of *NFE2* like *bZIP* transcription factor 2, obviously reduced TC, TG, and LDL-c levels by augmenting the expression of *ABCA1* [164]. Furthermore, the role of phagocyte-mediated efferocytosis effectively phagocytized and cleared apoptotic cells to attenuate atherosclerosis lesions [191]. In a recent study by Chen et al. [192], *ABCA1* was shown to be modulated by “find-me” (containing LPC) and “eat-me” (containing PtdSer, ANXA1, ANXA5, MEGF10, and GULP1) ligands to promote efferocytosis [192].

Interestingly, many traditional Chinese medicines are involved in *ABCA1* regulation and demonstrate efficacy against atherosclerosis. Yin-xing-tong-mai and Sini decoctions [193] increase *ABCA1* expression in macrophages by activating the *PPARγ–LXRα* pathway to attenuate atherosclerosis [194]. The Qing-Xue-Xiao-Zhi formula inhibits the *TLR4/MyD88/NF-κB* pathway to promote *ABCA1* expression [195]. Ethanol extract of Danlou tablet upregulates *ABCA1* by triggering the *PPARα* signaling pathway [196]. In apoE-/- mice, quercetin [197] and semen celosiae [198] have been found to promote *ABCA1* expression to protect against atherosclerosis. Chinese herbal compounds “Xuemai Ning” [199] and “Xinnaokang” [200] and flavonoids compounds [201] can up-regulate the expression of *ABCA1*. Curcumin can promote cholesterol efflux, reduce intracellular lipid content, and promote foam cell formation through the *miR–125a-5p/SIRT6* axis to overexpress *ABCA1* in macrophages [202]. In summary, previous studies have suggested that the upregulation of *ABCA1* expression inhibits the development of atherosclerotic lesions.

### 7.3. Ischemia/Reperfusion and Ischemic Heart Disease

Ischemia-reperfusion injury (IRI) is a complex phenomenon that occurs in numerous traumatic injuries and diseases. A prominent feature of IRI is the abrupt interruption of blood supply (ischemia), followed by the recovery of blood supply and re-oxygenation (reperfusion) [203]. IRI often causes reversible cell dysfunction, local and remote tissue destruction, and multiple organ failures [204]. In the heart, reperfusion after ischemia successfully attenuates ischemic myocardial damage. However, this leads to irreversible detrimental effects [205]. A previous study suggests that IRI is the predominant pathological condition in cardiovascular diseases, including IHD [206]. Epidemiological studies have shown that a deficiency of plasma HDL-C is closely related to an increased risk of IHD [207,208]. The protective effect of HDL-C in IHD is mainly confirmed by its involvement in RCT and its anti-inflammatory effects. In a study by Laura et al. [209], reconstituted HDL showed pleiotropic effects to protect isolated rat hearts against IRI, including promotion of prostaglandin and reduction of tumor necrosis factor-alpha release [209].

*ABCA1* plays a crucial role in nascent HDL-particle formation, maturation, and catabolism. Furthermore, many studies have suggested that *ABCA1* functions in ischemia/reperfusion-induced cardiomyocyte injury. However, there is no clear evidence to identify the changes in *ABCA1* expression during the ischemia or reperfusion stage. Although evidence has pointed out the association between the risk of IHD and loss-of-function variations in *ABCA1* [207,210,211,212], conflicts exist regarding whether inherited low plasma HDL-C levels accelerate the risk of IHD [213]. In TD patients, there was no significant increased risk of IHD [162]. Similarly, in the *ABCA1* loss-of-function mutation carriers, no increased risk of IHD was found [213]. Several studies have uncovered potential mechanisms by which *ABCA1* participates in IRI and IHD. As reported, HDL-stimulated nitric oxide (NO), an endogenous regulatory molecule, is released to trigger ischemic preconditioning against IRI [214,215]. The underlying mechanism is *ABCA1* mediating the activation of the *Akt/ERK/NO* pathway in ECs [216]. In a myocardial IRI mouse model, *ABCA1* was downregulated by *miR-27a* through the upregulation *NF-**κB* signaling pathway [217]. These results will bridge the knowledge gap in the biology of *ABCA1* in IRI and IHD.

### 7.4. Myocardial Infarction

Myocardial infarction (MI) is a major cause of disability and mortality worldwide [218]. It is characterized by the interruption of myocardial blood flow and reduction of myocardial oxygen supply, which leads to ischemic myocardial necrosis [219,220]. According to the universal definition of myocardial infarction, there are five subtypes of MI [221]. Among them, type 1 and type 2 MIs are the most common types in clinical cases. The difference between these two types is their occurrence with or without obstructive coronary disease [222]. Coronary atherosclerosis is the primary cause of acute atherothrombotic atherosclerosis. During unstable periods, plaque rupture and activated inflammation in the vascular wall often occurs [218]. It is important to note that HDL particles have anti-inflammatory effects; possess antithrombotic properties; prevent the oxidation of low-density lipoproteins; modulate vasomotor tone; and may improve EC function, proliferation, and migration [223,224].

It has been established that *ABCA1* plays a critical role in HDL production and cholesterol efflux and lipid homeostasis maintenance. Epidemiological studies have shown that mutations and loss-of-function in *ABCA1* significantly decrease HDL-C levels and accelerate cardiovascular diseases risk [222]. Consistent with this view, a 36-year-old man with MI had a combined ABCA1 and ApoA-I deficiency [225]. Additionally, Subramaniam et al. reported a 41-year-old man with premature recurrent MI caused by the *ABCA1* gene mutation, who had moderately decreased serum HDL-C and protein C levels with increased homocysteine [226]. A 45-year-old woman presented with three mutations (c.3137C > A, c.4595A > G, and c.5097G > T) in the *ABCA1* gene with MI, undetectable HDL, and multiple episodes of angina [227]. Remarkably, the R219K polymorphism in the *ABCA1* coding region is associated with a high risk of MI [78]. Moreover, in a group of young male survivors of MI, three mutations in the *ABCA1* gene (I883M, R219 K, and -477C/T) were identified, and their influence on long-term prognosis was analyzed. Notably, not all *ABCA1* mutations are associated with the risk of MI. For example, in the general Japanese population, a polymorphism in the *ABCA1* promoter region, G(-273)C, significantly decreased HDL-C levels but had no significant effect on MI risk [228]. However, there is no evidence that *ABCA1* polymorphisms are associated with genetic susceptibility to MI [229].

According to previous studies, *ABCA1* has a protective effect against atherosclerosis. However, it has adverse effects on cardiac function following MI [230]. Interestingly, Kavita et al. observed no significant changes in *ABCA1* mRNA transcripts in acute myocardial infarction (AMI) peripheral blood mononuclear cells [231]. Tina et al. [232] showed that niacin significantly stimulates *ABCA1* transcription by repressing the cyclic AMP/protein kinase A pathway to improve survival after MI [232]. Additional research is required to determine the correlation between *ABCA1* and MI.

### 7.5. CHD

CHD is one of the leading causes of mortality and imposes a substantial financial burden on modern society. The primary cause of CHD is the obstruction of blood flow in the coronary artery due to atherosclerosis or thrombosis [233]. Although numerous studies have advanced our understanding of the relationship between triglycerides and CHD, additional evidence suggest that circulating cholesterol is one of the most important risk factors for atherosclerosis and CHD [234,235,236]. In the last few years, cholesterol-lowering strategies have resulted in a prominent decrease in the total mortality of CHD [237]. Thus, the underlying correlation between cholesterol and CHD warrants further investigation.

Few studies have indicated that prebeta-1 HDL (preβ-1-HDL) level is a solid independent positive risk factor for CHD [238,239,240,241]. preβ-1-HDL is a subtype of HDL that is mainly formed by ApoA-I containing two copies of ApoA-I per particle. As mentioned previously [25], *ABCA1* plays a critical role in cholesterol efflux from macrophages and in the development and progression of CHD. Moreover, preβ-1-HDL is regarded as the principal acceptor of cholesterol efflux via ABCA1 mediated RCT. Moreover, it appears to be a substrate for lecithin–cholesterol acyl transferase, which esterifies cholesterol and plays a central role in HDL metabolism [242]. Consistent with the functions of *ABCA1* in cholesterol homeostasis, numerous population and basic studies have shown that CHD is a vital complication of *ABCA1* deficiency.

A large genetic study has supported the role of *ABCA1* in CHD susceptibility. In familial hypercholesterolemia, a genetic disorder of the *ABCA1* mutation, statin treatment can reduce the risk of CHD [243]. In the past decade, numerous polymorphisms (rs146292819 [244], rs1800976 [245], rs2230806 [R219K] [246], rs4149313 [M8831I] [247], rs9282541 [R230C] [56], -565C/T [248], A1092G [M233V] [249], rs363717, rs4149339, and rs4149338 [250]) in the *ABCA1* locus were significantly associated with susceptibility to CHD. The R230C/*ABCA1* variant features both a reduction in HDL-C levels and a protective effect against CHD [57].

Both gene mutations in *ABCA1* and DNA methylation modifications lead to this mRNA transcription deficiency. Previous reports have revealed a relationship between *ABCA1* promoter region methylation and CHD risk [251,252,253,254,255]. Infante, et al. [256] reported that *ABCA1*, *TCF7*, *NFATC1*, *PRKCZ*, and *PDGFA* DNA are highly methylated in the CD4+ and CD8+ T cells of patients with acute coronary syndrome (ACS) using epigenome-wide analysis [256]. Notably, the most severe clinical manifestation of CHD is ACS [257]. Fang et al. [253] indicate that high methylation of the *ABCA1* promoter is associated with decreased *ABCA1* expression and HDL-C levels, as might be expected. However, there was no significant association between *ABCA1* promoter region methylation status and plasma lipid concentration in an Iranian population [253]. Despite this, acetylsalicylic acid has been shown to attenuate *ABCA1* DNA methylation levels and protect against CHD [254]. These studies have revealed that epigenetic modifications might be a potential mechanism for CAD, and ongoing studies are needed to clarify these mechanisms. As summarized in this review, we conclude that *ABCA1* plays a role in a broad array of cardiovascular diseases, including dyslipidemia, atherosclerosis, ischemia/reperfusion, ischemic heart disease, myocardial infarction, and CHD, with different preventive effects (Figure 3).

## 8. Conclusions and Future Directions

*ABCA1* is a critical molecule involved in cholesterol metabolism and HDL production. Abnormalities in *ABCA1* gene expression or post-translational modifications of ABCA1 often lead to the excessive intracellular accumulation of cholesterol. As mentioned above, post-translational modifications of ABCA1 are closely related to its functions, including distribution, transport, degradation, and stabilization. However, the underlying mechanism of ABCA1 PTMs has not been identified in previous studies and thus remains to be elucidated in the future. In spite of the many studies that have identified that ABCA1 PTMs are involved in a variety of pathophysiological processes, few studies show the direct associations between the PTMs of ABCA1 in cardiovascular diseases. Therefore, it is critical to uncover the role of ABCA1 PTMs in cardiovascular diseases. It is of particular interest that researchers have had a breakthrough in progress toward identifying the Cryo-EM structure of human ABCA1. The structural observation developed provides us with a mechanistic understanding of disease mutations and lays a basis for the development of targeted drugs.

A series of landmark discoveries resulted in the development of the ‘HDL hypotheses’ and an inverse correlation between HDL-C concentration and cardiovascular diseases [255]. HDL particles have multiple functions and play important roles in promoting excess cholesterol efflux from macrophages to prevent lesions in arterial wall vessels [256]. Therefore, the elevation of circulating HDL levels by small HDL apoprotein-related mimetic peptides is a promising approach for the development of anti-atherogenic and anti-inflammatory drugs [257]. In addition, *ABCA1* is involved in HDL biogenesis. However, the other major clinical problem is that there is no specifical drug and synthetic ligand to regulate *ABCA1* expression. Therefore, there is an urgent need to find and develop targeted drugs that specifically regulate *ABCA1* expression, which will be the focus of future research.

Lipid accumulation and vessel wall inflammation are the two fundamental hallmarks of cardiovascular diseases [258]. To date, there has been conclusive evidence that *ABCA1* is involved in inflammation. Thus, the function of *ABCA1* in suppressing inflammation in macrophages should also be discussed. Literature data from animal models to humans indicate that macrophage-specific *ABCA1* deficiency is related to accelerated inflammatory cytokine release and pro-inflammatory gene expression [25]. The mechanisms of *ABCA1* suppression of inflammation involve a large number of signaling pathways, including Janus kinase 2 [259,260], Ca^2+^ [261,262], Rho family G protein cell division cycle 42 [263,264], and protein kinase A pathways [139]. Determining how *ABCA1* interacts with these inflammation pathways in cardiovascular diseases may ultimately uncover novel methods applied in cardiovascular disease therapy.

Much evidence supports the concept that macrophages play a critical role in the pathogenesis of various cardiovascular diseases, including, but not limited to, the formation of foam cells, proliferation in atherosclerotic lesions, necroptosis, and macrophage polarization [265]. Considering that *ABCA1* is most abundant in macrophages and its function is to maintain cholesterol homeostasis, much more research is needed to explore the underlying molecular mechanism by which *ABCA1* modulates cardiovascular diseases through macrophages. Additionally, it may shed light on the diagnosis and treatment of cardiovascular diseases.

## Figures and Tables

**Figure 1 jpm-12-01010-f001:**
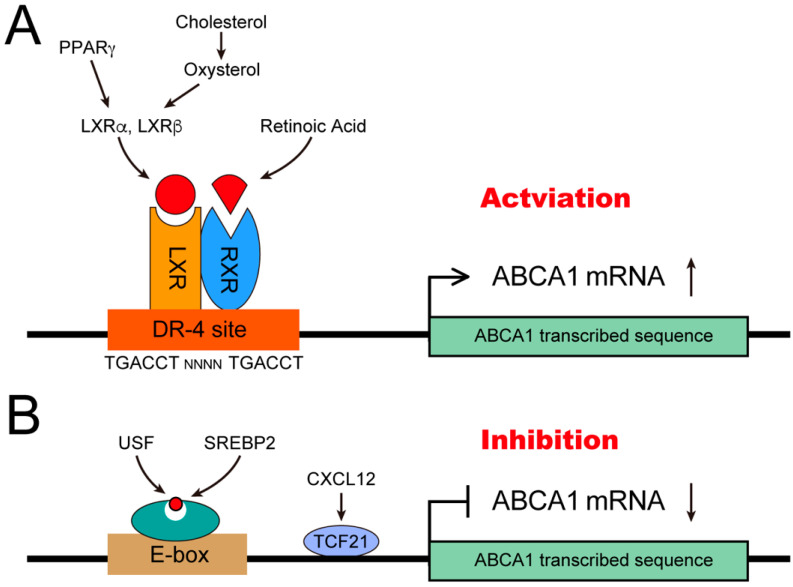
*ABCA1* expression is regulated by transcription factors. (**A**) PPARγ, oxysterol, and retinoic acid target LXR and RXR, respectively, to activate *ABCA1* expression. LXR and RXR bind to DR-4 element sequence, which are constituted by direct repeats of TGACCT and separated by four base-pairs. (**B**) USF and SREBP2 bind to E-box of *ABCA1*, and CXCL12 promotes TCF21 to interact with *ABCA1* promoter to inhibit *ABCA1* expression.

**Figure 2 jpm-12-01010-f002:**
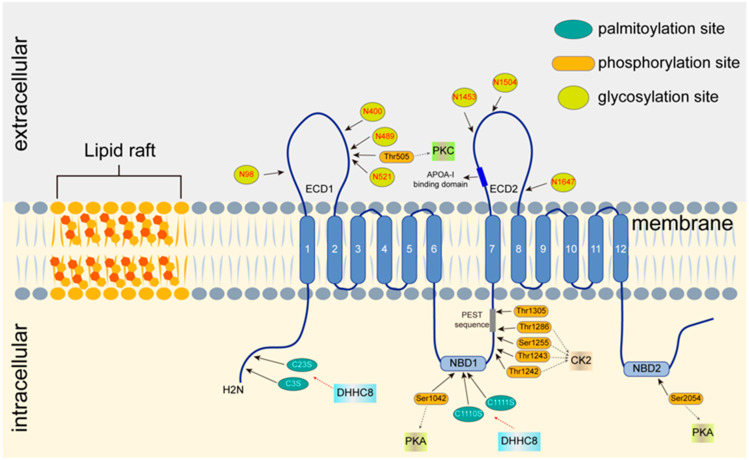
Plasma membrane location and post-translational modifications of ABCA1. According to the sucrose equilibrium density gradient, the plasma membrane was sub-divided into 10 fractions from low to high, the lipid raft region was fractions 1–5 and the non-lipid raft region was fractions 7–10. ABCA1 is located in the non-lipid raft region of the plasma membrane [86]. There were two palmitoylation sites in each of the N-terminus and NBD1 regions of ABCA1: C3S, C23S, C1110S, and C1111S. ABCA1 is palmitoylated by palmitoyl transferase DHHC8. Seven N-linked glycosylation sites are located in two ECD regions of ABCA1: N98, N400, N489, N521, N1453, N1504, and N1647. On the ECD1 region of ABCA1, Thr505 is phosphorylated by PKC. Ser1042 and Ser2054, which are phosphorylated by PKA, are located in NBD1 and NBD2 regions of ABCA1, respectively. Five additional phosphorylation sites in NBD1 are phosphorylated by CK2. NBD, nucleotide-binding domain; DHHC, Asp-His-His-Cys; ECD, extracellular domain; PKA, protein kinase A; PKC, protein kinase C; CK2, casein kinase 2.

**Figure 3 jpm-12-01010-f003:**
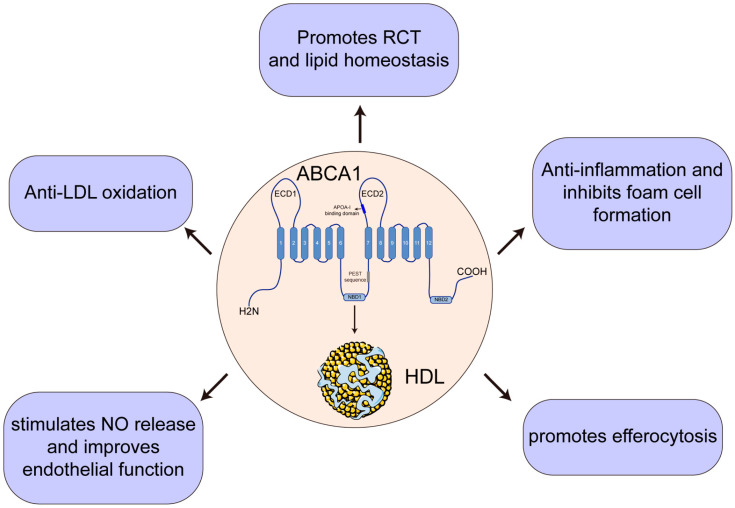
The preventive effects of *ABCA1* involvement in cardiovascular disease. RCT, reverse cholesterol transport; LDL, low-density lipoprotein; NO, nitric oxide.

## Data Availability

All of the 265 references are accessible in PubMed.

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
