# Peer review of "Role of ABCA1 in Cardiovascular Disease"

_jpm, 2022, doi:10.3390/jpm12061010_

Round 1

Reviewer 1 Report

This is a well-researched review article that focuses on the role of ABCA1 in cardiovascular disease, particularly in regulating cholesterol homeostasis. The authors first described the structure of ABCA1, followed by post-translational modifications of the protein that regulate its protein functions. Then, the authors discussed the dysregulation of ABCA1 in cardiovascular diseases. Overall, the manuscript was well-written and easy to follow and understand without redundant information in each section. My only comment is the lack of underlying signaling mechanisms by which ABCA1 may regulate cholesterol homeostasis. Having this section will help the readers to understand the significance of proper regulation of ABCA1 protein in mediating the downstream signaling events that regulate cholesterol homeostasis.

Author Response

Response to Reviewer 1 Comments:

Point 1: This is a well-researched review article that focuses on the role of ABCA1 in cardiovascular disease, particularly in regulating cholesterol homeostasis. The authors first described the structure of ABCA1, followed by post-translational modifications of the protein that regulate its protein functions. Then, the authors discussed the dysregulation of ABCA1 in cardiovascular diseases. Overall, the manuscript was well-written and easy to follow and understand without redundant information in each section. 

Response 1: Thank you very much for carefully reading our manuscript. We are encouraged by your interest in this manuscript.

Minor comment:
Point 2: My only comment is the lack of underlying signaling mechanisms by which ABCA1 may regulate cholesterol homeostasis. Having this section will help the readers to understand the significance of proper regulation of ABCA1 protein in mediating the downstream signaling events that regulate cholesterol homeostasis.

Response 2: Thank you for pointing this out and for your suggestion. Per your suggestion, we have added a section about the mechanisms of ABCA1 that regulate cholesterol homeostasis in the revised manuscript. Please refer to section 6.3, ‘Mechanisms of ABCA1 that regulate cholesterol homeostasis,’ on page 9, lines 371-380.

Reviewer 2 Report

The manuscript is a comprehensive description of the current knowledge of the role of ATP-binding cassette transporter A1 in cardiovascular disorders. 

I have some minor suggestions for the Authors.

1)    Figure 1= typing mistake: Activation 

2)    Please, indicate the period of the research in the literature and the used keywords on PubMed  

3)    Paragraph 2: SNPs are reported due to their association with cardiovascular risk; they are common polymorphisms in the general population. Are there differences among the different ethnicities? Is there some population with a higher risk for that?   

4)    Gender differences are known and reported in cardiovascular disorders. Please, add some comments about that, considering the topics discussed. 

5)    Adding comments about the existence of protective polymorphism related to the APO-I pathway, if there are, could be interesting  

6)    Line 463: Direction (typing error)

7)    Line 504: The Authors mention the need to complete more studies to understand the molecular mechanism of ATP-binding cassette transporter A1 in cardiovascular disorders: GWAS have been mentioned. Is it a unique type of approach to study? Which kind of approach could be used?

Author Response

Response to Reviewer 2 Comments:

Point 1: The manuscript is a comprehensive description of the current knowledge of the role of ATP-binding cassette transporter A1 in cardiovascular disorders. 

Response 1: Thank you very much for carefully reading our manuscript. We are encouraged by your interest in this manuscript.

Minor comments:
Point 2: Figure 1= typing mistake: Activation

Response 2: We thank the reviewer for their assistance in improving the quality of the manuscript and we apologize for the language errors. We amended the incorrect word in our revised manuscript – please refer to Figure 1 on page 3. Additionally, we revised the whole manuscript carefully to eliminate language errors. We believe that the language is now acceptable for the review process.

Point 3: Please, indicate the period of the research in the literature and the used keywords on PubMed 

Response 3: We appreciate the reviewer’s suggestion. Per your suggestion, we have indicated the period of the research (1975-2022) and the keywords used on PubMed in revision manuscript. Please refer to page 2, line 59-63.

Point 4: Paragraph 2: SNPs are reported due to their association with cardiovascular risk; they are common polymorphisms in the general population. Are there differences among the different ethnicities? Is there some population with a higher risk for that?  

Response 4: Yes, there are many differences among different ethnicities. According to previous studies, Caucasians have a higher cardiovascular risk. We appreciate the reviewer’s suggestion regarding this important issue. Per your request, we have added a section to discuss the association of gene polymorphism with cardiovascular risk in different ethnicities. Please refer to section 4, page 4, lines 154-163.

Point 5: Gender differences are known and reported in cardiovascular disorders. Please, add some comments about that, considering the topics discussed.

Response 5: We agree with the reviewer’s suggestion. In the revised manuscript, section 4 also discusses the differing association of gene polymorphism with cardiovascular risk between the sexes. Please refer to page 4, lines 163-175.

Point 6: Adding comments about the existence of protective polymorphism related to the APO-I pathway, if there are, could be interesting 

Response 6: Yes, there are several studies that report on protective/damage polymorphism related to the APOA-I pathway. Thank you for this valuable suggestion. We agree that the biological roles of protective polymorphism related to the APO-I pathway is potentially interesting. We have added a section that discusses this in the revision manuscript. Please refer to section 5, page 4, lines 176-187.

Point 7: Line 463: Direction (typing error) 

Response 7: We thank the reviewer for their assistance in improving the quality of the manuscript and we apologize for the language errors. We corrected this mistake in the revised manuscript – please refer to the Conclusion and future directions section, page 13, line 577. Additionally, we revised the whole manuscript carefully to eliminate language errors. We believe that the language is now acceptable for the review process.

Point 8: Line 504: The Authors mention the need to complete more studies to understand the molecular mechanism of ATP-binding cassette transporter A1 in cardiovascular disorders: GWAS have been mentioned. Is it a unique type of approach to study? Which kind of approach could be used?

Response 8: We appreciate the reviewer’s comment regarding this important issue and suggestion. GWAS means “genome-wide association study”. It is the most common and open approach to identify genetic variants associated with increase disease risk. All of the details are publicly available and accessible on the Internet (http://genetics.opentargets.org/). Hence, it makes users well-informed about disease-related sites, and researchers can prioritize them to assess their potential as targets for drug development. We therefore feel that GWAS is not a unique type of approach to study the molecular mechanism of ABCA1 in cardiovascular disorders.

Reviewer 3 Report

Journal

JPM (ISSN 2075-4426)

Manuscript ID

jpm-1759926

Type

Review

Title

Role of ABCA1 in cardiovascular disease

Authors

Jing Wang , Qianqian Xiao , Luyun Wang , Yan Wang * , Hu Ding * , Dao Wen Wang

Section

Mechanisms of Diseases

Special Issue

Cardiovascular Diseases—From Risk Factors to Diagnosis and Personalized Management

Date: 3 June 2022

In this review manuscript, Wang et al have discussed the role of ABCA1 in cardiovascular diseases. This is an interesting compilation of recent advances relevant to ABCA1, cholesterol metabolism in heart diseases. Here are my comments:

  1. For more emphasis on the family of this genes, authors can add the number of genes in this family and the reasons as to why this is relevant to the field as well as article. Refer the following paragraph "ATP-binding cassette transporter A1 (ABCA1) belongs to the ABC transporter super- 47 family and is located on chromosome 9q31.1. The length of the ABCA1 gene sequence is 48 149 kb, which contains 50 exons and 49 introns. ABCA1 is a 254 kD integral membrane 49 protein composed of 2261-amino-acid8 . ABCA1 is expressed in various tissues, including 50 the liver9 , intestine10, placenta11, pancreas12, lung13, heart14, macrophage15 and endothelial 51 cell16 . It also participates in numerous physiological and pathological processes17, includ- 52 ing inflammation18, cancer development19 , dysregulation of lipid metabolism20, type 2 di- 53 abetes mellitus21 and cardiovascular diseases 22. Although the human ABCA1 gene was 54 cloned in 1994, its biological function was not determined until 1999"
  2. Authors can mention that the current human genome annotation presents 49 ABC genes, arranged in seven subfamilies, designated A to G.  Refer "

Vasiliou, Vasilis, Konstandinos Vasiliou, and Daniel W. Nebert. "Human ATP-binding cassette (ABC) transporter family." Human genomics 3, no. 3 (2009): 1-10."

  1. Authors can discuss the findings of this study "Qian et al.37 reported on the discovery of cryo-EM structure of human ABCA1."
  2. Authors can improve the overall flow of their subsections by adding smaller introduction segments to individual subsections. For example, authors can add basic biology, relevance of discussed modification in molecular and clinical context. This will shed more light on the important biological contexture.
  • Authors can add a small introduction discussing post translational modifications in general.
  • Similarly, glycosylation, ubiquitination, phosphorylation and palmitoylation.
  1. Authors can expand the discussion of post translational modification of ABCA1 in cardiovascular diseases. Refer "Protein PTMs have been reported to be involved in the functional expression of ABC 147 transporters through a wide range of molecular mechanisms"
  2. Authors can improve the overall structure of the conclusion, for example rephrase the first sentence "Recently, it has become increasingly clear that cardiovascular diseases account for 464 one-third of all deaths in the world1 . "
  3. The conclusion and future directions has certain segments that needs to be moved to main text. For example the figure 3 and associated text.
  4. There is a lot of scope for improvement in the summary/concluding paragraphs. Authors should reconstruct key take-home points of the manuscript. The conclusion should crystallize the article and expand into future directions. The authors have compiled an interesting manuscript and if the authors read the manuscript again, the fine prints can be crystallized into a condensed, informative section.
  5. Smaller grammatical errors:
  • "Conclusion and Future Diections "
